# Fractional quantum ferroelectricity

Junyi Ji[1,2,4], Guoliang Yu [1,2,4], Changsong Xu[1,2] ✉ & H. J. Xiang [1,2,3] ✉

For an ordinary ferroelectric, the magnitude of the spontaneous electric polarization is at least one order of magnitude smaller than that resulting from the ionic displacement of the lattice vectors, and the direction of the spontaneous electric polarization is determined by the point group of the ferroelectric. Here, we introduce a new class of ferroelectricity termed Fractional Quantum Ferroelectricity. Unlike ordinary ferroelectrics, the polarization of Fractional Quantum Ferroelectricity arises from substantial atomic displacements that are comparable to lattice constants. Applying group theory analysis, we identify 27 potential point groups that can realize Fractional Quantum Ferroelectricity, including both polar and non-polar groups. The direction of polarization in Fractional Quantum Ferroelectricity is found to always contradict with the symmetry of the "polar" phase, which violates Neumann's principle, challenging conventional symmetry-based knowledge. Through the Fractional Quantum Ferroelectricity theory and density functional calculations, we not only explain the puzzling experimentally observed in-plane polarization of monolayer $\alpha$-In$_2$Se$_3$, but also predict polarization in a cubic compound of AgBr. Our findings unveil a new realm of ferroelectric behavior, expanding the understanding and application of these materials beyond the limits of traditional ferroelectrics.

Ferroelectricity, which is characterized by a reversible spontaneous polarization through application of an electric field, not only is important in fundamental physics, but also finds a wide range of applications, such as piezoelectric sensors/actuators[1–3] and non-volatile memory[4–6]. In common cases, the polarization originates from small atomic displacements, leading to the ordinary ferroelectricity, as illustrated in Fig. 1a. In such case, the direction of the polarization is consistent with the symmetry of the ferroelectric phase, e.g., the polarization has to be along the two-fold axis of a system with $C_{2v}$ symmetry. This is in accord with the well-known Neumann's principle[7], which states that the symmetry elements of any physical property of a crystal must include all the symmetry elements of the point group of the crystal.

In contrast to the small atomic displacements of ordinary ferroelectricity, recent studies report novel ferroelectricity with large atomic displacements that are comparable to lattice constants[8–13]. For example, the Cu ions are found to successively migrate through van der Waals layers in CuInP$_2$S$_6$[9,13], implying the polarization there can be very large and exhibits different values with different switching voltage. Regarding such large displacements, one may recall the modern theory of polarization (MTP), which is developed to resolve the multi-value problem of polarization when considering periodic boundary condition in computations. In MTP, the concept of polarization quantum, $\mathbf{Q} = \frac{e}{\Omega}\mathbf{a}$, where $e$ is the unit charge, $\Omega$ is the volume of the unit cell and $\mathbf{a}$ is a lattice vector, denotes the polarization caused by a unit charge traveling through the unit cell. With such understanding, the aforementioned systems actually exhibit ferroelectricity with an integer quantized polarization, dubbed as the quantum ferroelectricity (QFE, see Fig. 1b).

However, recent experimental and computational works[14–20] report unexpected large in-plane component of polarization in monolayer $\alpha$-In$_2$Se$_3$, which possesses $C_{3v}$ symmetry and allows only

[1]Key Laboratory of Computational Physical Sciences (Ministry of Education), Institute of Computational Physical Sciences, State Key Laboratory of Surface Physics, and Department of Physics, Fudan University, Shanghai 200433, China. [2]Shanghai Qi Zhi Institute, Shanghai 200030, China. [3]Collaborative Innovation Center of Advanced Microstructures, Nanjing 210093, China. [4]These authors contributed equally: Junyi Ji, Guoliang Yu. ✉e-mail: csxu@fudan.edu.cn; hxiang@fudan.edu.cn

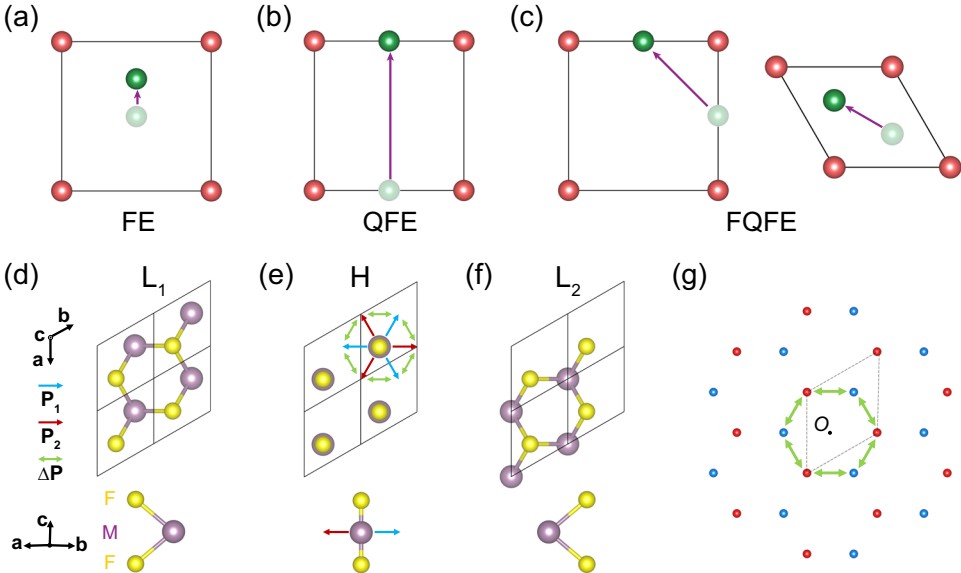

**Fig. 1 | Concept of FQFE.** Schematics of **a** FE, **b** QFE and **c** FQFE, supposing the ion with ±1 charges. The green and red balls represent movable ions and ligand ions, respectively. Top and side views of the FQFE example: **d** low-symmetry phase $L_1$, **e** high-symmetry phase H, **f** low-symmetry phase $L_2$. The black border indicates the unit cell. F includes two are fixed atoms (layers) F, and M is a movable atom (layer). The blue and red arrows depict the atomic displacements of M from H to $L_1$ and $L_2$, respectively. The green arrows show **ΔP**, the atomic displacements of M between $L_1$ and $L_2$. Since only M moves, the blue, red, and green arrows can also represent

$P_1$(polarization of $L_1$), $P_2$(polarization of $L_2$), and **ΔP** (polarization difference between low symmetry phases), respectively. **ΔP** cannot be invariant under a point symmetry operation ($C_{3z}$) of the low-symmetry phase, which leads to the FQFE. **g** The latticed form of $P_1$, $P_2$ and **ΔP**. The black dashed parallelogram depicts the "lattice" of polarization. The blue and red points represent $P_1$ and $P_2$, respectively. **ΔP** can be any vector between points with different color. Therefore, **ΔP** is non-zero and fractionally quantized, i.e. $\frac{1}{3}$**Q** along the [120] direction.

out-of-plane polarization accord to the Neumann's principle. Such contradiction indicates that the understandings of ordinary ferroelectricity and QFE are insufficient and new theory of ferroelectricity is highly desired. The development from ordinary FE to QFE is reminiscent of the series of Hall, quantum Hall and, fractional quantum Hall effects. It is thus promising to expect the existence of fractional quantum ferroelectricity (FQFE, see Fig. 1c). If it exists, one may wonder (i) how to identify FQFE by symmetry analysis, (ii) which fractional numbers to choose from, and (iii) how the Neumann's principle breaks down. Correctly answering such questions will help deepen the understanding of novel ferroelectricity.

In this work, we propose and demonstrate the existence of FQFE, through systematical group theory analysis and density functional theory (DFT) verifications. Our group theory analysis over all 230 space groups indicates that FQFE can exist in 27 point groups, of which 7 are polar while 20 are non-polar. It is also found that FQFE always contradicts with the Neumann's principle, i.e., the direction of polarization is not limited by the symmetry of ferroelectric phase, which is in strong contrast with ordinary FE. DFT calculations are further performed and demonstrate FQFE in monolayer α-In₂Se₃, cubic AgBr and many other systems.

## Results

### Framework of FQFE

The conceptual realization of the FQFE phenomenon in both tetrahedral and hexagonal systems is depicted in Fig. 1c. To elucidate the fundamental essence of FQFE, we discuss in detail a hexagonal system as an illustrative example. Within the FQFE, the ferroelectric behavior is exhibited through two distinct phases, denoted as $L_1$ and $L_2$, both adopting the MoS₂-type structural configuration. These phases, characterized by lower symmetry (see Fig. 1d, f), belong to space group *P-6m2* with the point group $D_{3h}$. Comprising both M and F ions, these phases involve the roles of M and F ions in the fractional polarization dynamics during the phase transition. Notably, M ions are mobile and contribute to the polarization switching, while the fixed F ions remain

stationary. To foster comprehension of the FQFE concept, we introduce an intermediary phase, H, which presents higher symmetry (as seen in Fig. 1e) and shares the composition of M and F ions with $L_1$ and $L_2$.

In the context of the M-F ion composition, $M_1$, $M_0$ ($M_0$ being the midpoint between $M_1$ and $M_2$), and $M_2$ symbolize the positions of M ions within $L_1$, H, and $L_2$, respectively. The polarization attributes of $L_1$ and $L_2$, along with the polarization difference between them, are ascribed to $P_1 \sim M_1$-$M_0$, $P_2 \sim M_2$-$M_0$, and **ΔP** $\sim M_2$-$M_1$, as indicated by the blue, red, and green arrows in Fig. 1e. Note that **P** denotes dipole moment here, rather than conventional polarization. The MTP indicates multi-values for $P_1$, $P_2$, and **ΔP**, which are visually illustrated through a lattice representation in Fig. 1g. It becomes evident that **ΔP** consistently maintains a non-zero in-plane value and lacks invariance under symmetry operations, such as the three-fold rotation around the z-axis ($c_{3z}$), belonging to the point group characterizing the low symmetry phases ($L_1$ or $L_2$). As we will find, **ΔP** is fractionally quantized besides the trivial integral lattice translation, which is protected by the $c_{3z}$ operation.

### Group theory analysis of FQFE

Firstly, we use group theory to present the basic FQFE framework and extract key features of FQFE. Then, we identify feasible space group-point group pairs that can realize FQFE realization in the case of one mobile atom. This framework is then generalized to the case of multiple mobile atoms. Finally, we address methods for assessing FQFE presence within a system and determining the other ferroelectric phase (e.g., $L_2$) for a given structure ($L_1$).

Let us discuss the framework in the language of group theory. The space group (and corresponding point group) of the structure F, as well as that of the low symmetry phases $L_1$ and $L_2$, are denoted as $G_F$ ($P_F$) and $G_L$ ($P_L$), respectively. $G_F$ can be either a symmorphic or non-symmorphic space group. For simplicity, we first consider the symmorphic case, in which the translational parts of all space group operations are lattice translations. In this case, the high symmetry

phase H can be constructed by setting $\mathbf{M}_0$ at the Wyckoff position with site symmetry $P_F$. Since M consists of a single atom, symmetry operations that maintain the invariance of $\mathbf{M}_0$ ($\mathbf{M}_{1,2}$) and F must be symmorphic space group operations within $G_F$. These operations collectively form the space group of H ($L_{1,2}$), which inherently constitutes symmorphic space groups. For simplicity, we consider only the point groups of H ($L_{1,2}$). Such point groups are the site symmetry groups of Wyckoff positions $\mathbf{M}_0$ ($\mathbf{M}_{1,2}$) within $G_F$, denoted as $P_F$ ($P_L$). $\mathbf{M}_1$ and $\mathbf{M}_2$ are attributed to different sites of the same Wyckoff position, since that the energies of $L_1$ and $L_2$ are degenerate. By contrast, if $G_F$ is a non-symmorphic space group, it becomes unfeasible to construct the H structure with the space group $G_F$, since there is only one mobile atom in the H structure. With above preparation, considering the scenario where a given space group $G_F$ features $\mathbf{M}_1$ and $\mathbf{M}_2$ (representing two coordinates of the identical Wyckoff position), the FQFE can be realized if the coordinate difference ($\mathbf{M}_2$-$\mathbf{M}_1$) involves fractional components such as 1/2 and 1/3.

Exploration of potential $G_F$-$P_L$ pairs for realizing FQFE with a single M atom follows these steps: (i) List the Wyckoff positions[21] of a given space group ($G_F$); (ii) Select a Wyckoff position with at least two distinct coordinates ($\mathbf{M}_1$ and $\mathbf{M}_2$), each exhibiting at least one fixed fractional component; (iii) Assess whether $\mathbf{\Delta P} \sim \mathbf{M}_2$-$\mathbf{M}_1$ possesses a fractional component; if so, FQFE is likely to occur, and the site symmetry group of $\mathbf{M}_{1,2}$ is $P_L$. Taken the non-symmorphic *Ccc2* space group (No.37) for an example: (i) The associated Wyckoff positions are listed in Table 1; (ii) It is found that Wyckoff positions 4b and 4c exhibit coordinates with fixed fractional components, while 4a and 8d do not; (iii) For Wyckoff position 4b, $\mathbf{\Delta P}$ between the two sites yields $(0,0,\square)$ and thus lacks a fractional component (component $\square$ is related to variables and is not considered in the FQFE frame); For Wyckoff position 4c, $\Delta P_y = 3/4\text{-}1/4 = 1/2$, facilitating FQFE with $P_L = c_{2z}$ (two-fold rotation along z). In such example, $\mathbf{\Delta P}$ pertains to the conventional cell, which is twice larger than the unit cell, due to that *Ccc2* is bottom centered with fractional translational symmetry of (1/2,1/2,0). In such case, the atom M shifts from (1/4,1/4,z) to (1/4,3/4,z+1/2) and such movement is different from (1/2,1/2,0), rendering $\mathbf{\Delta P}$ also being fractional within the unit cell.

Applying the above principle to all 230 space groups [14], it yields 571 $G_F$-$P_L$ pairs, which include 191 possible $G_F$ and 27 possible $P_L$, while do not include $C_1$, $C_6$, $C_{6v}$, $D_{6h}$, $O_h$ point groups, see Table S1. Interestingly, among the 27 possible $P_L$, 7 $P_L$ are polar point groups (e.g., $C_{3v}$) and 20 $P_L$ are non-polar (e.g., $T_d$). The latter type suggests that fractionally quantized polarization can exist in previously assumed non-polar systems, which is now brought back to the playground of ferroelectricity by FQFE. From Table S1, we can see that FQFE has $\mathbf{\Delta P} = \frac{n}{m}\mathbf{Q}$ (m = 2,3,4,6,8 and n is an integer) in the conventional cell, as well as being fractional within the unit cell. Notably, $\mathbf{\Delta P}$ is non-invariant under $P_L$ in all these cases, which is due to the fact that the fractional components of $\mathbf{M}_1$ and $\mathbf{M}_2$ are fixed and determined by the symmetries in $G_F$. Hence, such symmetry analysis indicates that FQFE always contradicts with Neumann's law.

For FQFE systems with multiple mobile atoms, the structure can also be decomposed into F and M, only the latter now possessing multiple atoms. Each atom within M, denoted as $M^i$, corresponds to the single movable atom in the previous simple case, resulting in fractionally quantized polarization $\mathbf{\Delta P^i}$. The cumulative polarization $\mathbf{\Delta P}$ is the sum of $\mathbf{\Delta P^i}$, which is also naturally being fractionalized. Notably, all $M^i$ atoms should maintain the site symmetry of the corresponding Wyckoff positions of the space group $G_F$. For instance, in a MXene[22–25] like material of $Sc_2CO_2$ [see SM], F comprises two Sc atoms and one C atom, with $G_F$ = *P-3m1*, while two O atoms in M share positions with site-symmetry $G_L(P_L)=P3m1(C_{3v})$. Each O atom contributes $\frac{1}{3}\mathbf{Q}$ and it yields the total $\mathbf{\Delta P} = \frac{2}{3}\mathbf{Q}$ [see SM].

Lastly, utilizing Table S1, we outline how to determine the presence of FQFE in a given low symmetry structure and how to identify the other symmetry-related low-symmetry phase(s). For the given low symmetry phase $L_1$, one divides $L_1$ into M and F, by assigning F with higher symmetries ($G_F$ is a supergroup of $G_L$). Each Wyckoff position of $M^i$ in $G_F$ should match one of those listed in Table S1 and have at least one fractional component choosing from $\{\frac{1}{2}, \frac{1}{3}, \frac{2}{3}, \frac{1}{4}, \frac{3}{4}, \frac{1}{6}, \frac{5}{6}, \frac{1}{8}, \frac{3}{8}, \frac{5}{8}, \frac{7}{8}\}$. If such a division is feasible, FQFE may be present. Then, apply $G_F$ symmetry operations that do not present in $G_L$ to $L_1$ to obtain other low symmetry phase(s), $L_2$. Note that applying different symmetry operations to $L_1$ might lead to different $L_2$.

## FQFE in monolayer α-In₂Se₃

Monolayer $\alpha$-$In_2Se_3$ has gained significant attention as a 2D ferroelectric material. It conforms to No.156 space group of *P3m1* (point group $C_{3v}$) with a three-fold rotation axis perpendicular to the monolayer, which indicates pure out-of-plane polarization according to Neumann's principle. However, intriguingly, monolayer $\alpha$-$In_2Se_3$ was predicted to exhibit not only out-of-plane polarization, but also a substantial in-plane component[14], which has been subsequently confirmed through experiments[17–19]. Such observed in-plane polarization behavior in $\alpha$-$In_2Se_3$ contradicts Neumann's principle, raising questions about the novel ferroelectricity and its underlying physics.

Applying the FQFE theory, we proceed to construct the ferroelectric phase $L_2$ from the initial state $L_1$ in monolayer $\alpha$-$In_2Se_3$. $L_1$ comprises five atomic layers arranged as Se-In-Se-In-Se, as depicted in Fig. 2a. $L_2$ can be constructed following these steps: (i) Dividing $L_1$ into M and F, with the latter involving the upper two layers and the lower two layers, i.e., Se-In---In-Se. Besides other symmetries in $G_L$, F further exhibits inversion symmetry $I$, leading to $G_F = G_L + IG_L = P\text{-}3m1$ (No. 164). M is represented by the middle Se atom, occupying Wyckoff position 2d within $G_F$ (see Table S2), which is a case listed in Table S1. (ii) The application of inversion, a symmetry operation in $G_F$ but absent in $G_L$, on $L_1$ yields $L_2$, as depicted in Fig. 2a. Since $G_F$ is a symmorphic space group, high symmetry phase H (see Fig. 2b) can be constructed by placing M, i.e. the middle layer Se atom, at Wyckoff positions 1a or 1b, each possessing site symmetry $P_F = D_{3d}$. The polarization switching process, illustrated in Fig. 2a, involves the middle Se atom moving from being directly above the lower In atom to being situated directly below the upper In atom. This in-plane displacement yields $\frac{1}{3}\mathbf{a} + \frac{1}{3}\mathbf{b}$

**Table 1 | Wyckoff positions of space group *Ccc2* (No.37)[21]**

| Multiplicity | Wyckoff letter | Site symmetry | Coordinates (0,0,0) + (1/2,1/2,0) + |
|---|---|---|---|
| 8 | d | $C_1$ | $(x,y,z)(-x,-y,z)(x,-y,z+1/2)(-x,y,z+1/2)$ |
| 4 | c | $C_2$ | $(1/4,1/4,z)(1/4,3/4,z+1/2)$ |
| 4 | b | $C_2$ | $(0,1/2,z)(0,1/2,z+1/2)$ |
| 4 | a | $C_2$ | $(0,0,z)(0,0,z+1/2)$ |

The "(0,0,0) + (1/2,1/2,0)" below "Coordinates" indicates that the conventional cell is twice the size of the unit cell and "(1/2,1/2,0)" in the conventional cell is a lattice vector in the unit cell. Therefore, only the listed coordinates are taken into consideration.

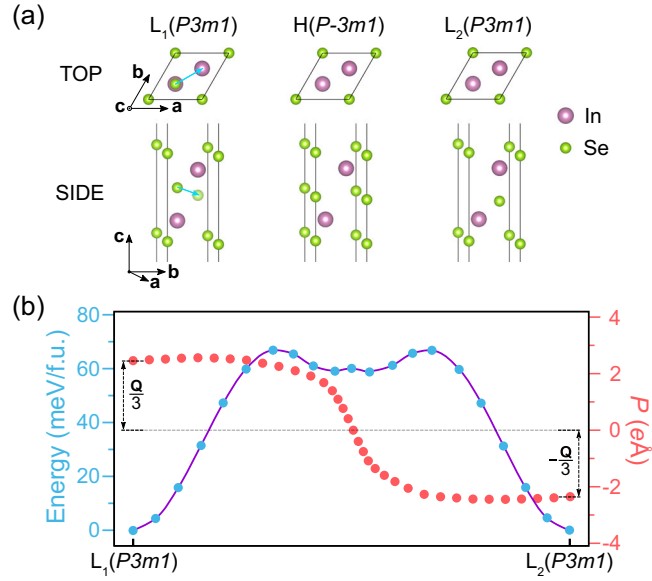

**Fig. 2 | Structure and ferroelectricity of monolayer α-In₂Se₃. a** Top and side views of monolayer α-In₂Se₃, corresponding to the L₁, H, and L₂ phases, respectively. $\frac{1}{3}\mathbf{a} + \frac{1}{3}\mathbf{b}$, the in-plane displacement of M (Se) in the ferroelectric phase transition is shown by the blue arrow. **b** NEB calculation of the energy barrier and evolution of the polarization intensity along the path similar to the one in Ref. 14. The amplitude represents the in-plane polarization magnitude along the [110] direction. The positive and negative signs indicate the polarization toward [110] and the [−1−10] direction, respectively. Here the polarizations of L₁ and L₂ are $\mathbf{P}_1 = \frac{1}{3}\mathbf{Q}$ and $\mathbf{P}_2 = -\frac{1}{3}\mathbf{Q}$, respectively. **Q** is the polarization quantum along the [110] direction. The polarization difference is $\mathbf{P} = \frac{2}{3}\mathbf{Q}$.

and consequently generates a non-zero in-plane fractionally quantized polarization, which contradicts the *P3m1* symmetry of the system. According to such analyses, the observed in-plane polarization of monolayer α-In₂Se₃ actually arises from the presently proposed FQFE.

DFT calculations are further performed to verify the contradiction between the in-plane polarization of monolayer α-In₂Se₃ and its symmetry. The minimum energy pathway between L₁ and L₂ is determined using the climbing image nudged elastic band (CI-NEB) method[26,27]. The energy barrier is determined to be 68 meV/f.u. (see Fig. 2b), which is consistent with the value reported in Ref. 23. Our focus then shifts to the in-plane polarizations of monolayer α-In₂Se₃, which are assessed using the Berry phase approach[28–30]. As illustrated in Fig. 2b, the magnitude of in-plane polarizations of monolayer α-In₂Se₃ exhibits continuous variation along the L₁-L₂ pathway. Specifically, the in-plane polarizations of L₁ and L₂ phases along the [110] direction yield 2.37 and −2.37 *e*Å per unit cell, respectively, in agreement with previous results [15]. The polarization quantum $\mathbf{Q} = \frac{e}{\Omega}\mathbf{a}$ along the [110] direction is 7.11 *e*Å/unit-cell, where **a** represents the lattice vector along the [110] direction. Therefore, the DFT-derived in-plane polarization is $\frac{1}{3}\mathbf{Q}$, which is consistent with the FQFE theory. The polarization calculations demonstrate that monolayer α-In₂Se₃ indeed behaves as a ferroelectric with in-plane polarizations, a phenomenon at odds with the symmetry of *P3m1*. The polarization values from DFT calculations further verify the FQFE in α-In₂Se₃.

## FQFE in AgBr

Applying the FQFE theory, we proceed to predict the existence of FQFE within a non-polar point group. From Table S1, we choose $P_L = T_d$ that exhibits the highest symmetry, and the corresponding symmophic space group $G_F = Fm\text{-}3m$ (No. 225). Illustratively, the zinc blende structure (space group *F-43m*) is a typical crystal structure with $T_d$ symmetry. As shown in Fig. 3a, it possesses chemical formula AB, where A locates at (0,0,0) and B at (1/4,1/4,1/4). Either A or B can

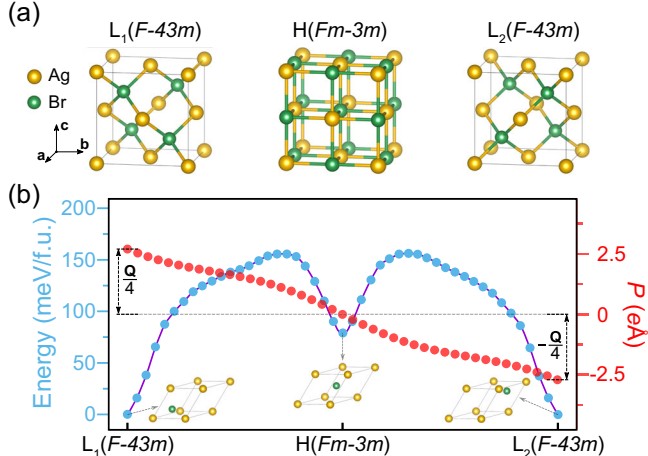

**Fig. 3 | Structure and ferroelectricity of bulk AgBr. a** Schematic structure of AgBr during the L₁-H-L₂ phase transition. The L₁,₂ phase belongs to *F-43m* and the H phase belongs to *Fm-3m*. The process can be considered as Br moving along the path $\left(\frac{1}{4},\frac{1}{4},\frac{1}{4}\right) \rightarrow \left(\frac{1}{2},\frac{1}{2},\frac{1}{2}\right) \rightarrow \left(\frac{3}{4},\frac{3}{4},\frac{3}{4}\right)$. **b** The energy barrier calculated by NEB and the evolution of the polarization in the primitive cell along the path in (a), where the primitive cells of L₁, H, L₂ are depicted. Here the polarizations of L₁ and L₂ are $\mathbf{P}_1 = \frac{1}{4}\mathbf{Q}$ and $\mathbf{P}_2 = -\frac{1}{4}\mathbf{Q}$, respectively. **Q** is the polarization quantum along the [111] direction. The polarization difference is $\mathbf{P} = \frac{1}{2}\mathbf{Q}$.

represent M (or F) atoms, with $G_F = Fm\text{-}3m$. Here, we designate A as F and B as M. The application of a $G_F$ operation that is not presented in $G_L$, such as inversion centered at the origin, to L₁ (L₂) results in the formation of the other low symmetry phase L₂ (L₁). The construction of H involves placing M at Wyckoff positions with site symmetry $P_F = O_h$, i.e. (1/2,1/2,1/2). Notably, H adopts a rocksalt structure with space group *Fm-3m*. During the transition from L₁ to L₂, as depicted in Fig. 3a, atom B shifts from (1/4,1/4,1/4) to (3/4,3/4,3/4). This displacement introduces a fractional shift of (1/2,1/2,1/2), rendering a fractionally quantized polarization. Notably, the lower symmetry phases L₁ and L₂ that exhibit polarizations actually belong to non-polar $T_d$ symmetry, showing the novelty of FQFE.

To investigate the above FQFE phenomenon further, we explore the Materials Project database[31] for materials featuring both *F-43m* and *Fm-3m* phases. Among these materials, AgBr emerges as our choice. Its ground state is a conventional rocksalt structure with space group *Fm-3m*[32–34]. Other materials exhibiting both *F-43m* and *Fm-3m* phases, such as ZnX (X = O, S, Se) and AlX (X = P, As, Sb), are listed in the supplementary material. Then, we turn to DFT calculations to exam the ferroelectricity of AgBr. By evaluating phonon spectra and employing molecular dynamics simulation, we confirm that the low symmetry *F-43m* phase is dynamically stable and exhibits good thermal stability at room temperature (see Fig. S1). The lattice constants of the high and low symmetry phases measure 5.84 and 6.31 Å, respectively. The energetically degenerate ground states of the system manifest as the L₁,₂ phases. Notably, the L₁,₂ phase is 74 meV/f.u. lower in energy than the H phase.

The demonstration of ferroelectricity in AgBr hinges on showcasing the presence of a switchable spontaneous polarization. Figure 3a portrays the kinetic pathways connecting the L₁ and L₂ phases. Given that the high symmetry phase H exists as an energy local minimum, the L₁-L₂ pathway can be deconstructed into two steps: L₁-H and H-L₂. During the L₁-H transition, the L₁ phase transforms into the H phase through the displacement of the Br atom from (1/4,1/4,1/4) to (1/2,1/2,1/2) along the [111] direction, overcoming an energy barrier of 155 meV/f.u. The subsequent H-L₂ transition involves the H phase transforming into another low symmetry phase (L₂ phase), as the Br atom shifts from (1/2,1/2,1/2) to (3/4,3/4,3/4) along the same direction,

necessitating an energy barrier of 76 meV/f.u. The $L_1$-$L_2$ polarization switching solely contends with the highest energy barrier, i.e., the 155 meV/f.u. barrier during the initial step. This energy barrier is similar to those observed in traditional bulk ferroelectric materials like $PbTiO_3$ and $BiFeO_3$[35], indicating that the fractionally quantized polarization of AgBr is likely to be switchable in experiments.

Figure 3b shows the evolution of polarization across the $L_1$-$L_2$ pathway for the AgBr primitive cell. The MTP is adopted to calculate polarization. Evidently, the system's polarization in the primitive cell experiences continuous changes as Br atoms move from (1/4,1/4,1/4) to (3/4,3/4,3/4). The polarizations of the $L_1$ and $L_2$ phases along the [111] direction are determined to be 69.80 and −69.80 μC/cm², respectively. According to MTP, the polarization quantum of AgBr system is 279.21 μC/cm² along the [111] direction, derived from $\mathbf{Q} = \frac{e}{\Omega}\mathbf{a}$, where $\mathbf{a}$ is the lattice vector along the [111] direction in the primitive cell. Consequently, both $L_1$ and $L_2$ phases display fractionally quantized polarization. Therefore, these findings establish AgBr as possessing spontaneous and switchable polarization, qualifying it as a material with FQFE in non-polar systems. This revelation underscores the potential for numerous other materials (such as those listed in Table S4) to harbor similar FQFE, thereby presenting opportunities for exploration and advancement across various technological domains. FQFE is also suitable for organic-inorganic systems, e.g. $NH_4Br$ with similar structures to AgBr [see SM].

## Discussion

According to MTP, electric polarization in a periodic system is multivalued. However, the difference in the polarization between two ferroelectric states is solely determined by the specific switching path. Taking $In_2Se_3$ as an example, though there are three possibilities in the direction of its polarization, it will be uniquely determined when applying a specific external field. Note that given the initial and final ferroelectric states, the polarization difference can only differ by an integer quantum polarization for different switching paths. Such a feature is in line with the "double-path" ferroelectrics discussed in Ref. 36.

It is well-known that ordinary ferroelectrics are piezoelectrics, pyroelectrics, and often ferroelastics. For QFE and FQFE materials with polar groups, they display piezoelectricity, pyroelectricity, and ferroelasticity, similar to ordinary FE materials. However, QFE and FQFE materials with non-polar groups generally lack piezoelectricity and pyroelectricity but may be ferroelastic as the lattice vectors in the two low-symmetry phases may be swapped [see SM]. This suggests that other direction-dependent properties (e.g., magnetic, transport, mechanical, optical properties) of FQFE materials are possible to be switched by electric fields.

In conclusion, we propose the concept of FQFE, which leads to polarization not only in polar systems but also in non-polar ones. A distinctive attribute of FQFE lies in its fractionally quantized polarization component. Our group theory analysis suggests the potential existence of FQFE in 27 point groups. Through the application of the FQFE theory, we explain the ferroelectric behavior in the monolayer α-$In_2Se_3$. Furthermore, employing our theory and first-principles calculations, we predict the FQFE within AgBr of the non-polar $T_d$ (*F-43m*) phase. The discovery of FQFE significantly expands the scope of ferroelectrics, opening avenues for exploring their properties and potential applications across diverse fields.

## Methods

### DFT calculations

Density functional theory (DFT) calculations are performed using the Vienna ab initio simulation package (VASP)[37]. The employed exchange and correlation functional adopts the generalized gradient approximation (GGA) as parametrized by Perdew, Burke, and Ernzerhof

(PBE)[38]. The adopted pseudopotentials are constructed using the projected enhanced wave (PAW) method[39,40]. The plane-wave energy cutoff is set to 500 eV. The structures are fully relaxed until the residual force on each atom is less than 0.01 eV/Å. The energy convergence criteria is set to $10^{-6}$ eV. The Brillouin zone is sampled using a Gamma-centered scheme with a $15 \times 15 \times 1$ $k$-point mesh for $In_2Se_3$ system and a $9 \times 9 \times 9$ k-point mesh for AgBr. For the $In_2Se_3$ monolayer, a 20 Å vacuum space is used and dipole corrections are adopted between adjacent layers to avoid interactions between neighboring periodic images. The energy barrier between different polarization states is calculated using the climbing image nudged elastic band (CI-NEB) method[41]. To examine the dynamical stability of AgBr, we calculate the phonon spectrum using Phonopy[42] with a $2 \times 2 \times 2$ supercell. To demonstrate the thermal stability of the FQFE systems, we perform ab initio molecular dynamics simulations for AgBr and monolayer $In_2Se_3$ systems at 300 K using $3 \times 3 \times 3$ and $4 \times 4 \times 1$ supercells, respectively.

## Data availability

The authors declare that all data supporting the findings of this study are available from the corresponding author upon request.

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

## Acknowledgements

The authors acknowledge financial support from the National Key R&D Program of China (No. 2022YFA1402901), NSFC (No. 11825403, 11991061, 12188101, 12174060, and 12274082), the Guangdong Major Project of the Basic and Applied Basic Research (Future functional materials under extreme conditions--2021B0301030005), and Shanghai Pilot Program for Basic Research—FuDan University 21TQ1400100 (23TQ017). C.X. also acknowledges support from the Shanghai Science and Technology Committee (Grant No. 23ZR1406600).

## Author contributions

H.X. and C.X. proposed the concept. J.J. conceived the framework and theory analysis. G.Y. carried out the calculations. J.J. and G.Y. prepared the initial draft of the manuscript. H.X. and C.X. supervised and guided the project from idea to framework design, theory analysis, and computations. All authors discussed the results and contributed to the revision of the manuscript.

## Competing interests

The authors declare no competing interests.
