## [Peer Review File · Nature Communications]

REVIEWER COMMENTS

Reviewer #1 (Remarks to the Author):

In this theoretical/computational contribution, the authors propose a new type of ferroelectricity, the Fractional Quantum Ferroelectricity (FQFE). Compared to the common understanding of ferroelectricity, the FQFE exhibits unexpected novel features: (i) the atomic displacements are much larger and are fractional times of lattice constants, and (ii) the direction of resulted polarization is always in contradictory with Neumann's principle. Such features make the concept of FQFE very intriguing. The establishment of FQFE is through rigorous and systematic group theory analysis, which not only indicates that FQFE can even exist in non-polar systems but also leads to a comprehensive table where all possible symmetries required by FQFE can be found. Examples of FQFE, monolayer In_2Se_3 , and bulk AgBr , are then provided and examined using DFT. The present work on FQFE is a breakthrough in the ferroelectricity theory and should be interesting to a broad audience from both experiments and theory. I thus would like to recommend its publication in Nature Communications.

Below are some comments for the authors' consideration, in order to further improve the manuscript.

1. The ferroelectric phase transition in a FQFE system involves ion displacements that are larger than those in a FE system, by more than an order of magnitude. Will it be more difficult for FQFE systems to undergo a ferroelectric phase transition compared to ordinary FE systems?
2. Ordinary ferroelectrics have been observed a long time before the present concept of FQFE. Is it due to the FQFE being rare? If it only exists in the examples discussed in the manuscript, the significance of this work can be reduced.
3. Please check the quotation of figures in "The application of inversion, a symmetry operation in GF but absent in GL, on L1 yields L2, as depicted in Fig. 2(c)", which seems to be a mistake.

Reviewer #2 (Remarks to the Author):

The manuscript from Ji et.al proposed the concept of "Fractional Quantum Ferroelectricity" which differs from the traditional Neumann's principle-based criteria used to identify ferroelectricity. The authors systematically confirmed this conclusion through space group analysis and provided two illustrative examples (e.g. In_2Se_3 and AgBr). In terms of scientific novelty, this manuscript should aim to achieve a level comparable to that of Nature Communications. However, I recommend that revisions be made before acceptance.

1. The authors highlighted that significant atomic displacement, often induced by weak interactions like van der Waals forces in periodic systems, can induce FQFE (Ferroelectric Quantum Ferroelectric Effect). However, it's crucial to recognize that significant displacement may destabilize bulk phase materials or incur a substantial energetic penalty. Therefore, I recommend that the authors address the thermal-stability analysis of FQFE materials in their discussion.
2. A technological concern was raised regarding the paraelectric phase of In_2Se_3 and AgBr , which displays metallic behavior. This behavior presents a challenge for the application of the Berry-phase method, which is typically used for insulators. How did the authors model the polarization curve in light of this issue? Did they employ a method involving the addition

or subtraction of integer polarization quanta to describe the overall polarization evolution? Please provide a detailed explanation, especially regarding AgBr, which has a non-polar point group.

3. Is FQFE suitable for molecular ferroelectric materials mediated by weak interactions? The authors should discuss an example to illustrate this.

4. Another concern is that the study focused solely on single-atom movements, while in actual conditions, multiple atoms often move simultaneously, resembling phonon modes.

5. Moreover, computational detail (e.g. functional, energy convergence, other corrections, et.al.) regarding DFT calculations should be further refined. The computational details provided in the supporting information should not duplicate those already mentioned in the main text.

Reviewer #3 (Remarks to the Author):

Please see the attachment.

In this manuscript, the authors proposed a new type (or a new perspective for classification) of ferroelectric material (fractional quantum ferroelectrics) and also explained a protocol to identify them by symmetry analysis. Such a new class of ferroelectrics contradicts, at first glance, the symmetry argument. However, experiments have clearly shown the existence of in-plane polarization in α -In₂Se₃. However, the three-fold rotation indicates the polarized direction should only be along the out-of-plane direction. The authors use symmetry analysis and DFT calculations to explain the puzzling experimental measurement, establish the concept of FQFE, and predict the polarization in cubic AgBr. The findings are very intriguing and have the potential to deepen our understanding of traditional ferroelectrics.

The paper is very well written and clear. The topic is appropriate for Nature Communications as it brings a major breakthrough to materials science in general through the new concept of FQFE. Hence, I strongly support its publication in Nature Communications.

Here are a few comments I want to discuss with the authors, and hope it could help improve the manuscript.

1. Without ligand ions to indicate the relative displacements, the illustration of polarization in Fig 1. (a-c) could be misleading to some audiences. For example, (a-c) could also mean shifting of unitcell origin, leading to no polarization.
2. Ferroelectric polarization is actually defined by its switching path (experimentally, the generated current that is measured during the switching). Near this switching path, the chosen reference structure is also very important. Relative to different reference structures, positive polarization according to one could be negative according to another, see arXiv:2204.06999. The switching path is experimentally chosen by the used external electric field.

Taking α -In₂Se₃, for example, the polarization orientation is actually defined by how the external fields are applied:

(Note that these two are not two energy minimal but exactly the same crystal structure.)

Why is this important? Because when talking about symmetry without considering the external electric field, and saying that C_{3v} forbids in-plane polarization is not at all contradicted with Neumann's principle in this materials, because the polarization directions are ill-defined without an electric field, for instance:

where you can see that C_{3v} (origin at any of the black dots) symmetry results in zero in-plane "polarization". Though this is not the context of this manuscript, this example explains Neumann's principle is not breaking, because it is the electric field symmetry + C_{3v} in total allows the in-plane polarization. (You may same any material that can be polarized in an external electric field, but mind that ferroelectric materials should be switchable.)

3. Another comment is about the reference structure. Choosing different reference structures could result in different signs of polarization (switching path is associated with the reference).

The H phase in the manuscript should result in different signs of polarization compared to the one discussed in other literature, e.g., the paraelectric phase in Fig. 2 (a) of Nat. Commun., 8, 14956 (2017), where the Se atom is chosen at the middle point between the shortest connection line among the nearest two In atoms. This corresponds to different switching paths. I am wondering if it is necessary to use the reference structure defined in Nat. Commun., 8, 14956 (2017) to calculate the same result as in Fig. 2 (b) of the manuscript. Comparing which one is the easiest switching path (lowest energy barrier) can be useful for the audience.

Comments and Replies

Reviewer #1

In this theoretical/computational contribution, the authors propose a new type of ferroelectricity, the Fractional Quantum Ferroelectricity (FQFE). Compared to the common understanding of ferroelectricity, the FQFE exhibits unexpected novel features: (i) the atomic displacements are much larger and are fractional times of lattice constants, and (ii) the direction of resulted polarization is always in contradictory with Neumann's principle. Such features make the concept of FQFE very intriguing. The establishment of FQFE is through rigorous and systematic group theory analysis, which not only indicates that FQFE can even exist in non-polar systems but also leads to a comprehensive table where all possible symmetries required by FQFE can be found. Examples of FQFE, monolayer In_2Se_3 , and bulk AgBr , are then provided and examined using DFT. The present work on FQFE is a breakthrough in the ferroelectricity theory and should be interesting to a broad audience from both experiments and theory. I thus would like to recommend its publication in Nature Communications.

Below are some comments for the authors' consideration, in order to further improve the manuscript.

Reply: We thank the reviewer very much for his/her high evaluation of our work and below valuable comments that improve our manuscript.

1. The ferroelectric phase transition in a FQFE system involves ion displacements that are larger than those in a FE system, by more than an order of magnitude. Will it be more difficult for FQFE systems to undergo a ferroelectric phase transition compared to ordinary FE systems?

Reply: Switching of polarization in FQFE materials can be easily achieved, though the ion displacements associated with FQFE systems are larger than ordinary FE systems. For example, the energy barrier to switch polarization of In_2Se_3 ($E_B = 68$ meV/f.u) is several times lower than typical ferroelectrics, such as PbTiO_3 ($E_B \approx 200$

meV/f.u.) [Nature 358, 136–138 (1992)] and LiNbO₃ ($E_B = 259$ meV/f.u.) [Phys. Rev. B 93, 134303 (2016)]. Moreover, the switching of polarization in In₂Se₃ has been experimentally achieved, which testifies the FQFE theory. Furthermore, Table S4 shows that there can be a good number of FQFE candidates, each of which exhibits low transition barrier. Such facts thus indicate the FQFE systems with large ion displacements can possibly be switched as easy as ordinary FE materials.

2. Ordinary ferroelectrics have been observed a long time before the present concept of FQFE. Is it due to the FQFE being rare? If it only exists in the examples discussed in the manuscript, the significance of this work can be reduced.

Reply: It is true that presently the FQFE systems are much fewer than ordinary FE materials. This is probably due to that FQFE is in violation of the Neumann's principle and thus leads to such phenomenon being overlooked. People can easily ignore the potential ferroelectricity when the point group is not one of the ten polar ones.

Without the symmetry limitations, the newly proposed FQFE mechanism can largely expand the number of ferroelectrics. As shown in Table S1, the FQFE can potentially be achieved with 28 point groups among all 32 crystal point groups, which is much more than the typical 10 polar point groups.

Actually, our work indeed predicts different types of potential FQFE candidate materials. As shown in the Supplementary Materials (SM), Table S4 collect 20 predicted candidates with the AgBr-type FQFE; Section S2 introduces FQFE in MXene material Sc₂CO₂; and Section S7 even shows the possibility of realizing FQFE in hybrid organic-inorganic materials. Such generality thus emphasizes the significance of our work, which expands the realm of ferroelectricity.

3. Please check the quotation of figures in “The application of inversion, a symmetry operation in G_F but absent in G_L , on L_1 yields L_2 , as depicted in Fig. 2(c)”, which seems to be a mistake.

Reply: We thank the Reviewer for pointing it out and now change “Fig. 2(c)” to “Fig. 2(a)” in main text, Page 7.

Reviewer #2

The manuscript from Ji et.al proposed the concept of "Fractional Quantum Ferroelectricity" which differs from the traditional Neumann's principle-based criteria used to identify ferroelectricity. The authors systematically confirmed this conclusion through space group analysis and provided two illustrative examples (e.g. In_2Se_3 and AgBr). In terms of scientific novelty, this manuscript should aim to achieve a level comparable to that of Nature Communications. However, I recommend that revisions be made before acceptance.

Reply: We thank the Reviewer for the nice evaluation of our work and the insightful comments reported below, addressing which further improves the quality of our manuscript.

1. The authors highlighted that significant atomic displacement, often induced by weak interactions like van der Waals forces in periodic systems, can induce FQFE (Ferroelectric Quantum Ferroelectric Effect). However, it's crucial to recognize that significant displacement may destabilize bulk phase materials or incur a substantial energetic penalty. Therefore, I recommend that the authors address the thermal-stability analysis of FQFE materials in their discussion.

Reply: It is commonly true that significant displacements are hard to occur and may lead to instability. However, as indicated in the main text and SM, the energy barriers to trigger polarization switching in FQFE materials can be as low as those in typical ferroelectrics (please see reply to comment 2 of Reviewer #1). Moreover, following the suggestion of the present reviewer, we have performed *ab initio* molecular dynamics (AIMD) simulations for the FQFE systems of In_2Se_3 and AgBr , for both the FE state and the intermediate state. As shown in Fig. R1, both example FQFE systems exhibit good thermal-stability at room-temperature. It thus indicates that the polarization switching in FQFE systems can be safely achieved. Note that the in-plane ferroelectricity (i.e., FQFE) in In_2Se_3 was demonstrated experimentally.

We follow the reviewer's suggestion and now add a new section of "6. The thermal-stability analysis of FQFE materials", as well as a new Fig. S4 [Fig. R1] in the

Fig. R1 Thermal stability of FQFE materials. (a) and (b) show the *ab initio* molecular dynamics (AIMD) simulations of AgBr for F-43m and Fm-3m phase (L and H phase), respectively, at 300 K. (c) and (d) show the AIMD simulations of monolayer α -In₂Se₃ for FE and fcc' phases at 300 K. The inset shows the corresponding structure after 5 ps of simulation.

2.A technological concern was raised regarding the paraelectric phase of In₂Se₃ and AgBr, which displays metallic behavior. This behavior presents a challenge for the application of the Berry-phase method, which is typically used for insulators. How did the authors model the polarization curve in light of this issue? Did they employ a method involving the addition or subtraction of integer polarization quanta to describe the overall polarization evolution? Please provide a detailed explanation, especially regarding AgBr, which has a non-polar point group.

Reply: It is true that the Berry-phase method is only applicable for insulating systems. In this work, we find that both In_2Se_3 and AgBr maintain insulating along the switching paths, and thus yield robust polarization values from Berry phase method. As shown in Fig. R2, the electronic band structures for both In_2Se_3 and AgBr exhibit well-defined gaps for the FE and intermediate phases, and the values of gaps are always larger than 0.7 eV during the switching. Furthermore, Fig. R3 shows the continuous polarization values obtained from the Berry phase method along switching paths for both systems, indicating that such method is succeeded at each data point.

Fig. R2 Band structures of monolayer $\alpha\text{-In}_2\text{Se}_3$ and bulk AgBr . (a) and (b) show band structures for FE (L_1 or L_2) and fcc' phases of $\alpha\text{-In}_2\text{Se}_3$, respectively. (d) and (e) show band structures for FE (L_1 or L_2) and PE (H) phases of AgBr , respectively. (c) and (f) show the band gap evolution of monolayer $\alpha\text{-In}_2\text{Se}_3$ and AgBr along the switching path, respectively.

Fig. R3 The evolution of polarization calculated using Berry phase method, for (a) monolayer α - In_2Se_3 and (b) AgBr along the switching path.

3. Is FQFE suitable for molecular ferroelectric materials mediated by weak interactions? The authors should discuss an example to illustrate this.

Reply: It is a good point to expand the FQFE concept to molecular ferroelectric materials. Yes, the FQFE mechanism is suitable for molecular ferroelectric materials mediated by weak interactions. Here, we provide an example to demonstrate FQFE in organic-inorganic materials mediated by weak interactions [see Fig. R4]. This hypothetical structure is constructed by replacing the Ag^+ ion by NH_4^+ . The fractional polarization is similar to that in AgBr .

We follow the reviewer's suggestion and now add a sentence "FQFE is also suitable for organic-inorganic systems, e.g. NH_4Br with similar structures to AgBr [see SM]" in main text (on Page 9), as well as a new section of "7. FQFE in molecular materials", as well as a new Fig. S5 [Fig. R4] in SM (on Page 24).

Fig. R4 Illustration of realizing FQFE in molecular material of NH_4Br , where the NH_4^+ takes the position of Ag in AgBr .

4. Another concern is that the study focused solely on single-atom movements, while in actual conditions, multiple atoms often move simultaneously, resembling phonon modes.

Reply: We agree with the Reviewer that in ferroelectrics multiple atoms can move simultaneously. As detailed in Sec. 2 of the SM, we take the MXene-like material Sc_2CO_2 as an example and illustrate the FQFE with two oxygen atoms in one unit cell move simultaneously. Moreover, as inspired by the previous comment from the present Reviewer, the molecular material of NH_4Br (Fig. R4) is also an example of FQFE with multi-atom movements (i.e., the NH_4^+ group).

5. Moreover, computational detail (e.g. functional, energy convergence, other corrections, et.al.) regarding DFT calculations should be further refined. The computational details provided in the supporting information should not duplicate those already mentioned in the main text.

Reply: We thank the Reviewer for valuable feedback regarding the computational details in our study.

In main text, Page 11, we now rewrite the “**DFT calculations.**” as “Density functional theory (DFT) calculations are performed using the Vienna ab initio simulation package (VASP) [37]. The employed exchange and correlation functional adopts the generalized gradient approximation (GGA) as parametrized by Perdew,

Burke, and Ernzerhof (PBE) [38]. The adopted pseudopotentials are constructed using the projected enhanced wave (PAW) method [39,40]. The plane-wave energy cutoff is set to 500 eV. The structures are fully relaxed until the residual force on each atom is less than 0.01 eV/Å. The energy convergence criteria is set to 10^{-6} eV. The Brillouin zone is sampled using a Gamma-centered scheme with a $15 \times 15 \times 15$ k-point mesh for In_2Se_3 system and a $9 \times 9 \times 9$ k-point mesh for AgBr. For the In_2Se_3 monolayer, a 20 Å vacuum space is used and dipole corrections are adopted between adjacent layers to avoid interactions between neighboring periodic images. The energy barrier between different polarization states is calculated using the climbing image nudged elastic band (CI-NEB) method [41]. To examine the dynamical stability of AgBr, we calculate the phonon spectrum using Phonopy[42] with a $2 \times 2 \times 2$ supercell. To demonstrate the thermal stability of the FQFE systems, we perform ab initio molecular dynamics simulations for AgBr and monolayer In_2Se_3 systems at 300 K using $3 \times 3 \times 3$ and $4 \times 4 \times 1$ supercells, respectively”. In SM, Page 21, we delete the section of “6. Details of density functional theory calculations” and update the section number.

Reviewer #3

In this manuscript, the authors proposed a new type (or a new perspective for classification) of ferroelectric material (fractional quantum ferroelectrics) and also explained a protocol to identify them by symmetry analysis. Such a new class of ferroelectrics contradicts, at first glance, the symmetry argument. However, experiments have clearly shown the existence of in-plane polarization in α -In₂Se₃. However, the three-fold rotation indicates the polarized direction should only be along the out-of-plane direction. The authors use symmetry analysis and DFT calculations to explain the puzzling experimental measurement, establish the concept of FQFE, and predict the polarization in cubic AgBr. The findings are very intriguing and have the potential to deepen our understanding of traditional ferroelectrics.

The paper is very well written and clear. The topic is appropriate for Nature Communications as it brings a major breakthrough to materials science in general through the new concept of FQFE. Hence, I strongly support its publication in Nature Communications.

Here are a few comments I want to discuss with the authors, and hope it could help improve the manuscript.

Reply: We thank the Reviewer for the nice evaluation of our work and the insightful comments reported below, addressing which further improves the quality of our manuscript.

1. Without ligand ions to indicate the relative displacements, the illustration of polarization in Fig 1. (a-c) could be misleading to some audiences. For example, (a-c) could also mean shifting of unitcell origin, leading to no polarization.

Reply: We thank the Reviewer for the valuable suggestion. We now add ligand ions to indicate the relative displacements [see Fig. R5].

We now add “The green and red balls represent movable ions and ligand ions, respectively” in the main text, and update Fig. 1(a-c) in main text (Page 13) and Fig. S3 in SM (Page 21).

Fig. R5 Schematics of (a) FE, (b) QFE and (c) FQFE, supposing the ion with ± 1 charges.

The green and red balls represent movable ions and ligand ions, respectively.

2. Ferroelectric polarization is actually defined by its switching path (experimentally, the generated current that is measured during the switching). Near this switching path, the chosen reference structure is also very important. Relative to different reference structures, positive polarization according to one could be negative according to another, see [arXiv:2204.06999](https://arxiv.org/abs/2204.06999). The switching path is experimentally chosen by the used external electric field.

Taking α -In₂Se₃, for example, the polarization orientation is actually defined by how the external fields are applied:

(Note that these two are not two energy minimal but exactly the same crystal structure.)

Why is this important? Because when talking about symmetry without considering the external electric field, and saying that C_{3v} forbids in-plane polarization is not at all contradicted with Neumann's principle in this materials, because the polarization directions are ill-defined without an electric field, for instance:

where you can see that C_{3v} (origin at any of the black dots) symmetry results in zero in-plane “polarization”. Though this is not the context of this manuscript, this example explains Neumann's principle is not breaking, because it is the electric field symmetry + C_{3v} in total allows the in-plane polarization. (You may same any material that can be polarized in an external electric field, but mind that ferroelectric materials should be switchable.)”

Reply: We thank the Reviewer for this valuable point about ferroelectric polarization. We agree with the Reviewer that the polarization difference between two states is determined by the switching path that is experimentally chosen by the used external electric field (see arXiv:2204.06999). In fact, as shown in our next reply, the polarization difference between two states in In_2Se_3 depends on the switching path. But given the two states (i.e., initial and final states), the polarization difference can only differ by an integer quantum polarization for different switching paths [see Fig. R7].

We agree with the Reviewer that the Fractional Quantum Ferroelectricity (FQFE) we discussed in this work does not volatile with the symmetry analysis as the polarization difference between two states is determined by the switching path. However, the usual Neumann's principle (i.e., the symmetry elements of any physical

property of a crystal must include all the symmetry elements of the point group of the crystal) that is suitable for describing ferroelectrics with small ionic displacements is no longer applicable to the FQFE case.

In the revision, we have added these discussions and cited arXiv:2204.06999. We add “According to MTP, electric polarization in a periodic system is multi-valued. However, the difference in the polarization between two ferroelectric states is solely determined by the specific switching path. Taking In_2Se_3 as an example, though there are three possibilities in the direction of its polarization, it will be uniquely determined when applying a specific external field. Note that given the initial and final ferroelectric states, the polarization difference can only differ by an integer quantum polarization for different switching paths. Such feature is in line with the “double-path” ferroelectrics discussed in Ref. [36].” in main text, Page 10.

3. Another comment is about the reference structure. Choosing different reference structures could result in different signs of polarization (switching path is associated with the reference). The H phase in the manuscript should result in different signs of polarization compared to the one discussed in other literature, e.g., the paraelectric phase in Fig. 2 (a) of Nat. Commun., 8, 14956 (2017), where the Se atom is chosen at the middle point between the shortest connection line among the nearest two In atoms. This corresponds to different switching paths. I am wondering if it is necessary to use the reference structure defined in Nat. Commun., 8, 14956 (2017) to calculate the same result as in Fig. 2 (b) of the manuscript. Comparing which one is the easiest switching path (lowest energy barrier) can be useful for the audience.

Reply: We thank the Reviewer for the valuable suggestion regarding the choice of the reference structure and its impact on the sign of polarization. It's important to acknowledge that selecting different reference structures can indeed lead to different signs of polarization, as the switching path is closely associated with the reference structure. To clarify such comment, we include a comparison among the results derived using different reference structures, including the PE phase, H phase, and their respective optimized phase transition paths [See Fig. R6]. One can find that the phase

in Fig. 2 (a) of [Nat. Commun., 8, 14956 (2017)] yields the most straightforward or energetically favorable switching path with the lowest energy barrier of 68 meV/f.u. Moreover, given the initial and final states, the polarization difference $\Delta P = P_1 - P_2$ can only differ by an integer quantum polarization for different switching paths [see Fig. R7].

We follow the review's suggestion and now add a new section of "8. Energy barrier and polarization difference along different switching paths for monolayer α -In₂Se₃.", as well as a new Fig. S6 [Fig. R6] and a new Fig.S7 [Fig. R7] in SM, Page 24.

Fig. R6 Nudged elastic band (NEB) calculation of the energy barrier for monolayer α - In_2Se_3 along switching paths with intermediate state (a) PE phase (b) fcc' phase, which corresponds to Fig. 2 (a) of [Nat. Commun., 8, 14956 (2017)], (c) H phase (d) fcc'' phase. The energy barriers are 850 meV/f.u., 68 meV/f.u., 1262 meV/f.u. and 625 meV/f.u. for (a), (b), (c), (d), respectively. The red arrows represent the movements of ions during the switching.

Fig. R7 The evolution of the polarization for monolayer α - In_2Se_3 in the primitive cell along the paths in Fig. R6: (a) PE phase (b) fcc' phase (c) H phase (d) fcc'' phase. \vec{Q} is the polarization quantum along the [110] direction. The polarization difference $\Delta P = P_1 - P_2$ are 4.74 eÅ , 4.74 eÅ , -9.48 eÅ and -9.48 eÅ per unit cell for (a), (b), (c), (d), respectively. The polarization of the gray points in panel (a) is determined by fitting other structures in this path due to their metallic behavior.

Summary of changes

The changes made are summarized below. Note that changes are highlighted in yellow in the main text and SM.

1. In response to the comment 3 of Referee #1, we made the following changes:

In main text, Page 7, we now change “Fig. 2(c)” to “Fig. 2(a)”.

2. In response to the comment 1 of Referee #2, we made the following changes:

In SM, Page 22, we now add a new section of “6. The thermal-stability analysis of FQFE materials” , as well as a new Fig. S4.

3. In response to the comment 3 of Referee #2, we made the following changes:

In main text, Page 9, we now add “FQFE is also suitable for organic-inorganic systems, e.g. NH₄Br with similar structures to AgBr [see SM].”.

In SM, Page 23, we now add a new section of “7. FQFE in molecular materials” , as well as a new Fig. S5.

4. In response to the comment 5 of Referee #2, we made the following changes:

In main text, Page 11, we now rewrite the “DFT calculations.” as “Density functional theory (DFT) calculations are performed using the Vienna ab initio simulation package (VASP) [37]. The employed exchange and correlation functional adopts the generalized gradient approximation (GGA) as parametrized by Perdew, Burke, and Ernzerhof (PBE) [38]. The adopted pseudopotentials are constructed using the projected enhanced wave (PAW) method [39,40]. The plane-wave energy cutoff is set to 500 eV. The structures are fully relaxed until the residual force on each atom is less than 0.01 eV/Å. The energy convergence criteria is set to 10⁻⁶ eV. The Brillouin zone is sampled using a Gamma-centered scheme with a 15 × 15 × 15 k-point mesh for In₂Se₃ system and a 9 × 9 × 9 k-point mesh for AgBr. For the In₂Se₃ monolayer, a 20 Å vacuum space is used and dipole corrections are adopted between adjacent layers to avoid interactions between neighboring periodic images. The energy barrier between different polarization states is calculated using

the climbing image nudged elastic band (CI-NEB) method [41]. To examine the dynamical stability of AgBr, we calculate the phonon spectrum using Phonopy[42] with a $2 \times 2 \times 2$ supercell. To demonstrate the thermal stability of the FQFE systems, we perform ab initio molecular dynamics simulations for AgBr and monolayer In₂Se₃ systems at 300 K using $3 \times 3 \times 3$ and $4 \times 4 \times 1$ supercells, respectively". In SM, Page 21, we delete the section of "6. Details of density functional theory calculations" and update the section number.

5. In response to the comment 1 of Referee #3, we made the following changes:
In main text, Page 13, we now add "The green and red balls represent movable ions and ligand ions, respectively." , and update Fig. 1(a-c).
In SM, Page 21, we update Fig. S3.
6. In response to the comment 2 of Referee #3, we made the following changes:
In main text, Page 10, we now add "According to MTP, electric polarization in a periodic system is multi-valued. However, the difference in the polarization between two ferroelectric states is solely determined by the specific switching path. Taking In₂Se₃ as an example, though there are three possibilities in the direction of its polarization, it will be uniquely determined when applying a specific external field. Note that given the initial and final ferroelectric states, the polarization difference can only differ by an integer quantum polarization for different switching paths. Such feature is in line with the "double-path" ferroelectrics discussed in Ref. [36]" .
7. In response to the comment 3 of Referee #3, we made the following changes:
In SM, we now add a new section of "8. Energy barrier and polarization difference along different switching paths for monolayer α -In₂Se₃." in Page 24, as well as a new Fig. S6 in Page 25 and a new Fig.S7 in Page 26.
8. In SM, Page 21, we now update Fig. S2 by painting Ag and Br atoms with the same color in Fig. 3 in the main text.

REVIEWERS' COMMENTS

Reviewer #1 (Remarks to the Author):

All the comments and concerns from last round of reviewing have been reasonably addressed, the quality of present work well meet the high standard of Nature Communications, it is recommended to publish as is

Reviewer #2 (Remarks to the Author):

After reviewing the revisions addressing my concerns, I believe the current manuscript is suitable for publication in Nat. Commun.

Reviewer #3 (Remarks to the Author):

The referee is grateful to the authors for engaging in thoughtful discussions and diligently addressing all comments. In this referee's assessment, the paper has been significantly improved and now stands as an excellent piece of work, deemed ready for publication.